# Long Non-Coding RNA-Ribonucleoprotein Networks in the Post-Transcriptional Control of Gene Expression

**DOI:** 10.3390/ncrna6030040

**Published:** 2020-09-17

**Authors:** Paola Briata, Roberto Gherzi

**Affiliations:** Gene Expression Regulation Laboratory, Ospedale Policlinico San Martino IRCCS, 16132 Genova, Italy

**Keywords:** long non-coding RNA 1, RNA binding protein 2, post-transcriptional regulation

## Abstract

Although mammals possess roughly the same number of protein-coding genes as worms, it is evident that the non-coding transcriptome content has become far broader and more sophisticated during evolution. Indeed, the vital regulatory importance of both short and long non-coding RNAs (lncRNAs) has been demonstrated during the last two decades. RNA binding proteins (RBPs) represent approximately 7.5% of all proteins and regulate the fate and function of a huge number of transcripts thus contributing to ensure cellular homeostasis. Transcriptomic and proteomic studies revealed that RBP-based complexes often include lncRNAs. This review will describe examples of how lncRNA-RBP networks can virtually control all the post-transcriptional events in the cell.

## 1. Introduction

Mammalian genomes are pervasively transcribed even though, in humans, only 19,000 proteins are coded for by less than 2% of the genome and, in the last two decades, it has become clear that the vast majority of the genome is transcribed as non-coding RNAs (ncRNAs) [1]. Long non-coding RNAs (lncRNAs), a largely underexplored class of ncRNAs arbitrarily classified as >200 nucleotides long, account for most of this pervasive transcription and more and more lncRNAs have been demonstrated to be functional molecules rather than transcriptional noise [1,2]. They are expressed in many different cell types and tissues at different levels, display strong cell- and tissue- specific expression, and are often characterized by poor conservation among species, at least at the primary sequence level [1,2]. Besides lncRNAs that display genomic features in common with protein coding-genes, others can be assigned to the following categories: (*i*) lncRNAs that are intergenic to protein-coding genes (lincRNAs); (*ii*) natural antisense transcripts (AS); and (*iii*) intronic lncRNAs [1,2]. In general, lncRNAs exhibit a surprisingly wide range of sizes, structural arrangements and functions and can be detected in the nucleus and/or the cytoplasm of expressing cells. All these features endow them with diverse and enormous functional potential even though they have also presented experimental challenges for their analysis [1,2].

Like proteins, lncRNAs exert their roles in all cell functions operating through different mechanisms. Their versatile features depend on several reasons but mainly on their subcellular localization and the adoption of specific structural modules with interacting partners, a process that may undergo dynamic changes in response to local cellular environments [3]. lncRNAs have been shown to be involved in diverse fundamental cellular processes such as proliferation and apoptosis, development and differentiation, X chromosome inactivation, and genomic imprinting [3]. They have also been implicated in human diseases such as coronary artery disease, amyotrophic lateral sclerosis, and Alzheimer’s disease [4,5,6] as well as in cancer with either oncogenic or tumor suppression functions [7]. LncRNAs can mediate their effects in *cis* or in *trans* by directly binding to DNA, RNA or proteins and can (*i*) influence the function of transcriptional complexes; (*ii*) modulate chromatin structures; (*iii*) regulate genome organization through interaction with nuclear matrix proteins; (*iv*) function as scaffolds to form ribonucleoprotein (RNP) complexes; (*v*) act as decoys for proteins and micro-RNAs (miRNAs) [2,3]. Thus, lncRNA-mediated control of gene expression may take place at transcriptional and/or post-transcriptional levels [2,3].

In general, lncRNAs interact with RNA-binding proteins (RBPs) that are conventionally viewed as proteins that bind to RNA through one or multiple RNA-binding domains and then change the fate or function of the bound RNAs [8]. A wide range of RBPs has been discovered and investigated over the years and proved to regulate gene expression at many levels but these are generally viewed as key players in post-transcriptional events [9,10]. The combination of the versatility of their RNA-binding domains with their structural flexibility enables RBPs to be involved in virtually all the post-transcriptional regulatory layers in the cell and to control the metabolism of a large array of transcripts [9,10]. RBPs establish highly dynamic interactions with other proteins, as well as with coding and non-coding RNAs, creating functional RNPs that regulate pre-mRNA splicing and polyadenylation, mRNA export, stability, localization and translation [9,10].

Excellent reviews are available on the roles of lncRNAs in transcriptional regulation and genomic organization. This review will focus on different levels of post-transcriptional control exerted by lncRNA/RBP interactions (*i*) polyadenylation and pre-mRNA splicing, (*ii*) mRNA export, (*iii*) mRNA decay, (*iv*) translation, (*v*) protein stability, (*vi*) miRNA maturation from precursors. We will not consider post-transcriptional effects dependent on base pairing between lncRNAs and other RNA species that do not involve RBPs.

## 2. LncRNAs, RBPs, and Regulation of pre-mRNA Processing

In order to produce a mature mRNA that can be efficiently translated into a protein, pre-mRNAs need extensive processing that can be recapitulated in (*i*) addition of cap structures at their 5′-end (capping), (*ii*) addition of stretches of A nucleotides at their 3′end (polyadenylation), and (*iii*) removal of introns with joining of exons (splicing). In certain circumstances, splicing and polyadenylation reactions can be modulated in order to originate two or more mRNA isoforms from a single pre-mRNA with processes defined as alternative polyadenylation (AP) and alternative splicing (AS) that concern more than 90% of intron-containing genes in humans [11,12]. The initial post-transcriptional modifications of pre-mRNA molecules—5′-end capping, splicing, and 3′-end formation by cleavage/polyadenylation—occur co-transcriptionally in the nucleus [13]. Indeed, seminal experiments performed in the early 2000s revealed that coupling early modifications of pre-mRNA with polymerase II-dependent transcription accelerates, by several orders of magnitude, the process of mRNA maturation [13]. Therefore, one could properly refer to these events as co- and post-transcriptional modification of nascent mRNAs. In recent years, a number of reports indicated that lncRNAs directly regulate AS events by utilizing three distinct modes: (*i*) the interaction with specific splicing factors (SFs) as well as with other SF-associated RBPs; (*ii*) the formation of RNA-RNA duplexes with pre-mRNA molecules [2,3], and (*iii*) the induction of chromatin remodeling that indirectly favors the AS of specific genes [2,3]. We will discuss here only the first mode of regulation.

Studies performed by the Chess laboratory in 2007 revealed that two abundant, predominantly nuclear lncRNAs, *MALAT1* (Metastasis Associated Lung Adenocarcinoma Transcript 1) and *NEAT1* (Nuclear Enriched Abundant Transcript 1), are associated with nuclear domains enriched in pre-mRNA splicing factors that are located in the interchromatin regions of the nucleoplasm of mammalian cells (speckles and paraspeckles) [14].

*MALAT1* co-localizes with several transcription factors as well as pre-mRNA processing factors and plays a critical role in coordinating transcriptional and post-transcriptional gene regulation [15]. Numerous RBPs (hnRNPH1, hnRNPK, hnRNPA1, hnRNPL, and PCBP1, just to mention a few) are required to ensure *MALAT1* proper localization to nuclear speckles [15]. Further, *MALAT1* has been described to interact with component of the pre-mRNA splicing complex (RNPS1, SRRM1, and AQR) as well as with a number of RBPs involved in specific pre-mRNA AS events (SRSF1, SRSF2, SRSF3, SON, hnRNPC, hnRNPH1, hnRNPL among others) [16,17]. Overall, *MALAT1* localizes to hundreds of genomic sites belonging to active genes, modulates the recruitment of splicing factors to a large number of actively transcribing loci, and its silencing severely affects pre-mRNA splicing in cultured cells [17,18,19,20,21]. Further, Prasanth and coworkers reported that *MALAT1* is able to modulate the phosphorylation status of the SF SRSF1 further reinforcing the notion that the lncRNA exerts a biological role as a coordinator of pre-mRNA splicing [17] (see also Section 5).

Kingston and coworkers have demonstrated that *MALAT1* colocalizes to many of its chromatin binding sites with another abundant lncRNA, *NEAT1,* even though the two lncRNAs display overall distinct binding patterns thus suggesting that they exert partly overlapping functions [20]. Interestingly, proteomic experiments revealed that both *MALAT1* and *NEAT1* interact with a common set of proteins that include the splicing factor ESRP2 and the scaffold protein SAFB2 that is involved in the regulated phosphorylation of SRSF1 by the kinase SAPK1 [20]. *NEAT1* is an exquisitely nuclear lncRNA and an essential structural component of paraspeckles that include the splicing factors SFPQ and NONO and control different aspects of gene expression [22]. Similar to *MALAT1*, also *NEAT1* recently proved to play an important role in modulating AS events. Shelkovnikova laboratory, taking advantage of a *Neat1* knockout mouse model, demonstrated that the lncRNA controls the AS of a group of genes important for neuronal proliferation and differentiation, cell–cell interactions in the central nervous system (CNS), synaptogenesis, and axon guidance [23]. Interestingly, *Neat1* also controls the AS of a group of RBPs including hnRNPA2B1, hnRNPH1, hnRNPD, hnRNPK, SRSF5, and SRSF7 [23]. *Neat1* knockout mice display a phenotype characterized by deficit in social interaction and rhythmic patterns of CNS activity [23]. Further evidence of the role of *Neat1* in regulating AS derived from a recent study that demonstrated the interaction of the lncRNA with the multifunctional RBP KHSRP. *Neat1*-KHSRP complex controls the process of metastatization of soft-tissue sarcomas by regulating AS events [24].

Another lncRNA localized to a nuclear compartment enriched in pre-mRNA splicing factors, is *Miat* (Myocardial Infarction Associated Transcript, a.k.a. *Gomafu*) that has been reported by Mattick and coworkers to be implicated in the pathogenesis of schizophrenia, a debilitating mental disorder affecting about 1% of the world population [25]. Authors demonstrated that *Miat* can regulate neuronal activity-dependent AS likely by acting as a scaffold for splicing factors (including SF1, SRSF1, and QK1) [25]. *Miat* transient downregulation that occurs upon neuronal depolarization allows the release of the splicing factors thus affecting AS events in neuronal cells [25].

A mass spectrometry-based analysis of molecular partners of *PANDAR* (Promoter Of *CDKN1A* Antisense DNA Damage Activated RNA)—a lncRNA involved in the regulation of proliferation and senescence whose overexpression has been observed in several human cancers and correlates with poor survival rate—allowed the identification of an unanticipated function of this lncRNA in modulating AS. Hennig and coworkers demonstrated that *PANDAR* interaction with PTBP1, a factor implicated in the regulation of AS events, results in modulated AS of *BCL2L1* pre-mRNA that encodes a potent inhibitor of cell death [26]. Authors hypothesize that *PANDAR* exerts a decoy function [26]. PTBP1 also interacts with *Pnky*, a neural-specific, nuclear lncRNA and modulates the expression and the AS of an overlapping set of transcripts [27]. Double knockdown experiments performed in neuronal stem cells indicate that the RBP and the lncRNA function in the same pathway [27].

The interaction of *LINC01133* with the SF SRSF6 proved to contribute to the ability of the lncRNA to modulate the Epithelial to Mesenchymal Transition (EMT) in colorectal cancers [28]. *LINC01133* is an abundant lncRNA whose expression is down-regulated upon colon cancer cell treatment with TGFβ, a potent inducer of EMT [28]. *LINC01133*-mediated inhibition of the SRSF6 function appears to be required for the lncRNA-mediated inhibition of EMT [28]. This observation supports the notion previously reported by our laboratory that TGFβ induces EMT by modulating the activity of RBPs involved in AS regulation [29].

By investigating the functions of *DSCAM-AS1* (Down Syndrome Cell Adhesion Molecule antisense 1)—a lncRNA overexpressed in invasive breast cancers—De Bortoli and coworkers reported that the lncRNA, besides affecting global gene expression and producing changes in the AS of its targets, influences polyadenylation by regulating the alternative 3′ UTR usage of 360 genes [30]. These changes in the early steps of the post-transcriptional regulation of gene expression appear to depend on the interaction between *DSCAM-AS1* and the nucleoplasm-enriched RBP hnRNPL [30].

## 3. LncRNAs, RBPs, and Regulation of mRNA Nuclear Export

Mature (capped, spliced, polyadenylated) mRNAs rapidly associate with RBPs and, together with various other RNA species (rRNA, tRNA, miRNA precursors, lncRNA), are transported from the nucleus to the cytoplasm through the nuclear pore complex (NPC) in the context of RNPs [31]. Despite the fact that mammalian cells synthesize a multitude of distinct mRNAs and that the composition of each individual RNP is unique and extremely dynamic throughout its life, export of the vast majority of mRNAs utilizes a single export receptor, the heterodimeric export receptor NXF1-NXT1 that mediates translocation through the NPC [31]. The export receptor is displaced at the cytoplasmic side of the NPC to release the RNPs into the cytoplasm. Directionality of the transport is controlled by distinct sets of DEAD-box ATPases that regulate RNPs association to and dissociation from the NXF1-NXT1 complex [31,32]. Importantly, mRNA nuclear export can undergo intense regulation by a variety of stimuli [32] that can also contribute to drug-induced eradication of cancer cells [33].

Recently, Prasanth and coworkers demonstrated that the overexpression of a predominantly nuclear lncRNA (*ROCR*, a.k.a. *LINC02095*) promotes breast cancer proliferation by facilitating the expression of the oncogenic transcription factor SOX9 [34]. *ROCR* favors both transcription and nuclear export of *SOX9* mRNA and its silencing in breast cancer cells reduces the cytoplasmic levels of *SOX9* mRNA [34]. Interestingly, SOX9 displays strong nuclear localization in highly invasive triple-negative breast cancer cells as opposed to other breast cancer subtypes [34]. Although nuclear retention of *SOX9* mRNA in cells depleted of *ROCR* is demonstrated, authors do not provide information on how the lncRNA affects the process of mRNA export and on the identity of the RBP(s) that, interacting with *ROCR*, contributes to its function.

Chromosome translocations may result in the exchange of DNA sequences between genes. Many such gene fusions are strong driver mutations in neoplasia and have provided fundamental insights into the pathogenetic mechanisms of certain tumors [35]. Chimeric mRNAs resulting from genomic rearrangements need to be translocated to the cytoplasm in order to be translated into the resulting oncogenic proteins [35]. Wang and coworkers recently reported on the involvement of the *MALAT1* in the regulation of nuclear export of chimeric mRNAs encoding the oncogenic fusion proteins PML-RARA, MLL-AF9, MLL-ENL, and AML1-ETO [36]. These authors show that nuclear export of the chimeric mRNAs depends on the *MALAT1* expression levels [36]. They propose a complex regulatory mechanism that involves the methylation of mRNAs to form N6-methyladenosine (m6A). m6A modification of mRNA accounts for the most abundant mRNA internal modification and has emerged as a widespread regulatory mechanism that controls gene expression in diverse physiological processes [37]. RBPs able to catalyze the m6A modification (writers), to recognize the m6A modification (readers), and to abrogate this specific modification (erasers) have been identified and characterized in recent years [37]. m6A has been reported to enhance mRNA export from the nucleus through the interaction of the m6A-modified mRNAs with the “reader” RBPs YTHDC1 and SRSF3 that function as adaptors for the NXF1-dependent mRNA export pathway [37]. Wang and coworkers provide evidence that *MALAT1*, upon interaction with oncogenic fusion proteins in nuclear speckles, promotes the interaction between the fusion proteins and the m6A methyltransferase cofactor METTL14 thus controlling the chimeric mRNA-exporting process through the m6A reader YTHDC1 [36]. The results of this study suggest the possibility that other lncRNAs, besides *MALAT1*, could provide a platform for the association of m6A “readers” with m6A-modified specific mRNAs to influence their nuclear export.

## 4. LncRNAs, RBPs, and Regulation of mRNA Decay

It is well known that the abundance of an mRNA is a function not only of its synthesis, processing, and nuclear export, but also of its degradation rate in the cytoplasm [38]. mRNA decay is an essential step in gene expression as it can rapidly set the levels of transcripts that undergo translation. A multitude of RBPs and/or non-coding RNAs can bind to specific elements of a certain mRNA and dictate its degradation rates via their ability to recruit (or exclude) the mRNA degradation machineries which perform the complex events of deadenylation, decapping and degradation of the RNA body [38]. Several cues can activate signal transduction pathways and modify the general mRNA decay machinery through their interaction with specific RBPs and this affects the mRNA decay rate and abundance [38]. We will describe and discuss here below examples of lncRNAs that contribute to the regulation of mRNA decay through their interaction with RBPs and, in turn, modulate important cellular functions and crucial pathological events.

An important example of lncRNA-RBP network operating in the cytoplasm and modulating the relevant cell function of maintaining genomic stability in human cells is based on the lncRNA *NORAD* [39,40]. *NORAD* (non-coding RNA activated by DNA damage) is highly conserved, broadly and abundantly expressed in mammalian cells and tissues, and induced after DNA damage [39,40]. Importantly, inactivation of *NORAD* triggers dramatic aneuploidy in previously karyotypically stable cell lines. In a search for *NORAD*-interacting proteins, Mendell and co-workers found that this lncRNA functions as a multivalent binding platform for the PUMILIO (PUM) family of RBPs, with the capacity to sequester a significant fraction of the cellular pool of PUM1 and PUM2 and, in turn, to limit their ability to repress target mRNAs [39]. RBPs of the PUM family bind with high specificity to sequences in the 3′ UTRs of target mRNAs and stimulate deadenylation and decapping, resulting in accelerated turnover and decreased translation [41]. Among PUM targets are a large set of factors that are critical for mitosis, DNA repair as well as DNA replication and their excessive repression in the absence of *NORAD* perturbs accurate chromosome segregation and can induce tetraploidization [39,40,41]. These findings have revealed a lncRNA-dependent mechanism that regulates a highly dosage-sensitive family of RBPs, uncovering a post-transcriptional regulatory axis that maintains genomic stability in mammalian cells and contributes to an emerging concept that a major class of lncRNAs function as molecular decoys. More recently, *NORAD*, whose sequence is characterized by several repetitive units, has been studied in order to identify additional interacting partners [42]. Ulitsky and coworkers found the RBP KHDRBS1 (a.k.a. SAM68) binds to *NORAD* and is required for *NORAD* function in antagonizing PUM [42]. This provides a paradigm for how repeated elements in lncRNAs synergistically contribute to complex tasks and for how a lncRNA can interact with multiple RBPs in order to operate a specific function.

Another lncRNA endowed with several distinct functions is *H19* [43]. In a systematic search to detect regulatory RNA species interacting with the RBP KHSRP in multipotent mesenchymal C2C12 cells, we identified, among others, *H19* [44]. We demonstrated that KHSRP directly interacts with *H19* in the cytoplasm of proliferating undifferentiated C2C12 cells and that this interaction favors the decay-promoting function of KHSRP on labile transcripts, such as *Myog*, through recruitment of the Exosome complex [44]. AKT activation during C2C12 differentiation induces KHSRP dissociation from *H19* and, as a consequence, *Myog* mRNA is stabilized whereas KHSRP is able to shuttle to nuclei where it promotes maturation of myogenic miRNAs from precursors, thus favoring myogenic differentiation (see also Section 6) [44]. In a sense, *H19* can be viewed as a modulator of two important and distinct post-transcriptional regulatory steps that lead to myogenic differentiation.

Recently, we identified a lncRNA expressed in epithelial tissues which we termed *Epr* (Epithelial cell Program Regulator, a.k.a. *BC030874*). *Epr* is rapidly downregulated by TGF-β and its sustained expression largely reshapes the transcriptome, favors the acquisition of epithelial traits, and reduces cell proliferation in cultured mammary gland cells as well as in an animal model of orthotopic transplantation [45]. Mechanistically, *Epr* interacts with chromatin and regulates the transcription of several genes [46] including the cyclin-dependent kinase inhibitor *Cdkn1a*. Interestingly, *Epr* changes *Cdkn1a* gene expression by affecting both its transcription and mRNA decay through its association with the transcription factor SMAD3 and the RBP KHSRP, respectively [45]. KHSRP is predominantly an mRNA decay promoting factor in this cellular context and the interaction with *Epr* blocks its ability to induce decay of *Cdkn1a* mRNA.

The lncRNA *LERFS* (Lowly Expressed in Rheumatoid Fibroblast-like Synoviocytes) is expressed at low levels in fibroblast-like synoviocytes (FLSs) derived from patients suffering for rheumatoid arthritis (RA) and regulates the migration, invasion, and proliferation of FLSs through interaction with the RBP SYNCRIP (a.k.a. hnRNPQ) [47]. Under healthy conditions, the *LERFS*-SYNCRIP complex, by binding to the mRNA of *RHOA*, *RAC1*, and *CDC42*—the small GTPase proteins that control the motility and proliferation of FLSs—, decreases the stability and/or translation of the target mRNAs and downregulates their protein levels [47]. In RA FLSs, decreased *LERFS* levels induce a reduction of the *LERFS*-SYNCRIP complex and this, in turn, reduces the binding of SYNCRIP to the target mRNAs thus increasing their stability or translation [47]. More specifically, *LERFS* and SYNCRIP regulate the stability and the translation of *RAC1* mRNA but regulate only the mRNA translation of *RHOA* and *CDC42* (see also Section 4) [47]. In general, these findings suggest that a decrease in synovial *LERFS* may contribute to the synovial aggression and joint destruction that are features of RA and targeting *LERFS* may have therapeutic potential in patients suffering for RA.

The lncRNA *UCA1* (Urothelial Carcinoma-Associated 1) has been found as a target of the CAPERα/TBX3 transcriptional repressor complex which is required to prevent premature senescence of primary cells, to regulate the activity of core senescence pathways in mouse embryos, and to control cell proliferation by repressing the transcription of *CDKN2A* gene (a.k.a. p16INK) and the RB pathway [48]. *UCA1* is a direct transcriptional target of CAPERα/TBX3 repression and its overexpression is sufficient to induce senescence [48]. In proliferating cells, hnRNPA1 binds and destabilizes *CDKN2A* mRNA whereas during senescence, *UCA1* sequesters hnRNPA1 and this, in turn, stabilizes *CDKN2A* mRNA [48]. Dissociation of the CAPERα/TBX3 co-repressor during oncogenic stress activates *UCA1* which, therefore, can be considered a tumor suppressor. See Section 4 for *UCA1*-dependent translational regulation and its opposite outcome in tumorigenesis.

Akiyama and colleagues demonstrated that *MYU* (MYC-Upregulated, a.k.a. *VPS9D1-AS1*) is a lncRNA transcriptionally induced by MYC upon its activation by the WNT signaling [49]. *MYU* is upregulated in most colon cancers and required for the tumorigenicity of colon cancer cells. Mechanistically, *MYU* associates with the RBP hnRNPK to stabilize *CDK6* mRNA and thereby promotes the G1-S transition of the cell cycle [49]. The authors also propose that hnRNPK and *MYU* hinder the inhibitory effect of miR-16 on *CDK6* mRNA [49]. Importantly, the WNT/MYC/*MYU*-mediated upregulation of CDK6 is essential for cell cycle progression and clonogenicity of colon cancer cells [49].

Another lncRNA playing a role in tumorigenesis is *LINC-ROR* (Regulator of Reprogramming) whose knockout in colon cancer cells suppresses cell proliferation and tumor growth. *LINC-ROR* plays an oncogenic role in part through regulation of *MYC* mRNA expression [50]. The lncRNA interacts with the RBPs PTBP1 (a.k.a. hnRNPI) and hnRNPD (a.k.a. AUF1) and is required for PTBP1 binding to *MYC* mRNA, while the interaction of *LINC-ROR* with hnRNPD inhibits its binding to *MYC* mRNA. As a result, *MYC* mRNA stability is increased and this leads to enhanced cell proliferation and tumorigenesis [50]. See also Section 4 for *LINC-ROR* functions in translation.

Cao and coworkers demonstrated that miR-1 promotes IFNG- (a.k.a IFN-γ) activated innate response in macrophages during *Listeria monocytogenes* infection through increasing the expression of *Stat1* mRNA [51]. From a mechanistic point of view, miR-1 targets the lncRNA *Sros1* (Suppressive non-coding RNA of STAT1) for degradation [51]. In noninfected macrophages *Sros1* blocks the interaction of *Stat1* mRNA with the RBP CAPRIN1 while the *Listeria monocytogenes*-induced degradation of *Sros1* releases CAPRIN1 that is made available to bind and stabilize the *Stat1* mRNA thus leading to increased STAT1 protein levels [51]. This ultimately strengthens IFNG signaling in the macrophages and promotes an innate immune response to intracellular bacterial infection.

## 5. LncRNAs, RBPs, and Translation Regulation

Translation is a multistep process comprising initiation, elongation, termination and ribosome recycling [52]. During initiation, the ribosome is recruited to the mRNA and scans the 5′ untranslated region of the transcript for the presence of the translation start codon. Under most conditions, initiation is the rate-limiting step of translation and therefore it is tightly regulated. Several key signaling pathways, including mammalian/mechanistic target of rapamycin (mTOR), mitogen activated protein kinases (MAPKs), and integrated stress response (ISR) pathways, converge on the initiation step to control the rate of protein synthesis in response to a variety of stimuli [52]. Control of mRNA translation plays a pivotal role in the regulation of gene expression in embryonic and adult tissues and defects in the translation process are deleterious for development and physiology [52]. During recent years, several lncRNAs have been identified as regulators of distinct steps of their target mRNA translation.

The lncRNA *TRERNA1* (Translational Regulatory, a.k.a. *treRNA*) was identified through genome-wide computational analysis [53]. *TRERNA1* is upregulated in breast cancer primary and lymph node metastasis samples and its expression stimulates tumor invasion in vitro and metastasis in vivo [53]. Authors found that *TRERNA1* downregulates the expression of the epithelial marker CDH1 (a.k.a. E-cadherin) by suppressing the translation of its mRNA and identified a novel RNP complex—consisting of the RBPs hnRNPK, FXR1, and FXR2 as well as the splicing factors PUF60 and SF3B3—that is required for *TRERNA1* function [53]. In more detail, PUF60-SF3B3 dimer interacts with hnRNP K, FXR1, and FXR2 to form a *TRERNA1*-containing RNP complex that, in turn, binds to eIF4G1 affecting translation [53].

Mo and coworkers have found that *LINC-ROR* is transcriptionally induced by TP53 (a.k.a. p53) and, at the same time, is a strong negative regulator of TP53-mediated cell cycle arrest and apoptosis [54]. Unlike MDM2 that causes TP53 degradation through the ubiquitin–proteasome pathway, *LINC-ROR* suppresses TP53 translation through direct interaction with the phosphorylated form of the RBP PTBP1 (a.k.a. hnRNPI) in the cytoplasm [54]. This suggests that the *LINC-ROR*-PTBP1-TP53 axis may constitute an additional surveillance network for the cell to better respond to various stresses (see also Section 3 for the role of *LINC-ROR* in mRNA decay control). The same group demonstrated that PTBP1 can also form a functional RNP with the lncRNA *UCA1* and increase the *UCA1* RNA stability [55]. In addition, in this case the phosphorylated form of PTBP1, predominantly in the cytoplasm, is responsible for the interaction with *UCA1* [55]. The interaction of *UCA1* with PTBP1 suppresses the protein level of CDKN1B (a.k.a. p27KIP1) by competitive inhibition, although the precise mechanism is still unclear. Authors demonstrate that the complex comprising *UCA1* and PTBP1, has an oncogenic role in breast cancer both in vitro and in vivo [55]. See Section 3 for *UCA1*-dependent regulation of mRNA stability and its opposite outcome in tumorigenesis.

*LncMyoD* (a.k.a. *1700025L06Rik*) is a lncRNA whose primary sequence is not well conserved between human and mouse models while its locus, gene structure, and function are preserved [56]. *LncMyoD* is transcribed next to the *Myod* gene and is directly activated by MYOD during myoblast differentiation. Knockdown of *LncMyoD* strongly inhibits terminal muscle differentiation, mainly due to an unsuccessful exit from the cell cycle [56]. Authors demonstrate that *LncMyoD* directly binds to the RBP IGF2BP2 (a.k.a IMP2) and negatively regulates IGF2BP2-mediated translation of genes able to modulate proliferation such as NRAS and MYC and this contributes to the failure of myoblast terminal differentiation [56].

Bozzoni and co-workers describe another regulatory circuitry controlled by a muscle-specific cytoplasmic lncRNA, *Lnc-Smart* (Skeletal Muscle Regulator of Translation, a.k.a. *Gm14635*), which is essential for proper differentiation of murine myogenic precursors [57]. By direct base pairing with a G-quadruplex region present in the *Mlx-γ* mRNA, *Lnc-Smart* prevents the translation of the mRNA by counteracting the activity of the RBP DHX36 endowed with RNA helicase function [57]. The time-restricted, specific effect of *Lnc-Smart* on the translation of *Mlx-γ* isoform modulates also the general subcellular localization of total MLX proteins (isoforms α and β), impacting on their transcriptional output and promoting proper myogenesis and mature myotube formation [57]. In more detail, *Lnc-Smart* depletion leads to alteration of the differentiation program with defects in myoblast fusion while its overexpression produces an apoptotic phenotype. Authors propose that *lnc-SMaRT* needs to be precisely controlled in time and quantity in order to fine-tune the balance between differentiation and apoptosis to ensure proper myogenesis [57].

The lncRNA *BCYRN1* (Brain Cytoplasmic RNA, a.k.a. *BC200*) regulates RNA metabolism in neural cells by modulating local translation in the postsynaptic dendritic microdomain by interacting with components of the translational machinery, such as eIF4A, eIF4B, and PABPC1 [58]. Lee and coworkers identified the RBPs hnRNPE1 and hnRNPE2 as *BCYRN1*-interacting proteins using a yeast three-hybrid screening. hnRNPE1 and hnRNPE2 bind to *BCYRN1* and can rescue the *BCYRN1*-dependent inhibition of translation by competing with eIF4A for binding to the lncRNA in an in vitro system [58].

## 6. LncRNAs, RBPs, and Post-Translational Modifications

Post-translational modifications occur in almost every protein during or after its translation and represent an extremely powerful tool operated by the cell in order to regulate the activity, stability, localization, interactions or folding of proteins by inducing their covalent linkage to new functional chemical groups, such as phosphate, acetyl, methyl, carbohydrate and ubiquitin [59]. Different post-translational modifications lead to distinct effects on target proteins and result in disparate biological consequences, from survival to apoptosis, from proliferation to differentiation, from activation to quiescence [59].

FUS (Fused in Sarcoma) is a multifunctional RBP that plays essential roles in post- transcriptional gene expression and possesses the ability to contribute to RNP granule formation via an RNA-dependent self-association [60]. FUS ability to interact with multiple RNA species accounts for its multiple functions. FUS (*i*) binds to nascent pre-mRNAs and acts as a molecular mediator between RNA polymerase II and RNAU1 small nuclear RNA-containing RNP thereby coupling transcription and splicing, (*ii*) binds to its own pre-mRNA and autoregulates its expression, and (*iii*) promotes homologous recombination during DNA double-strand break repair [60]. Numerous mutations in the *FUS* gene have been identified in patients suffering for two severe neurodegenerative disorders, amyotrophic lateral sclerosis and frontotemporal lobar degeneration [60]. Although the molecular mechanisms of FUS-dependent neurotoxicity are poorly understood, high concentrations of the RBP within RNA granules have been proposed to promote the formation of irreversible pathological aggregates [60]. Two recent papers point to lncRNA-dependent post-translational modifications of FUS as critical mechanisms affecting the cellular concentration and activity of the RBP and, in turn, its cellular functions. Nagai and coworkers reported that silencing of the *Drosophila* lncRNA *hsrω* converts FUS from a mono-to di-methylated arginine status via upregulation of the arginine methyltransferase 5 (PRMT5) [61]. PRMT5-dependent modification of FUS promotes its proteasomal degradation, thus leading to a strong downregulation of its cellular levels. Although in this case FUS regulation by the lncRNA is indirect, it is also interesting to note that *hsrω* interacts with and organizes a number of RBPs including TARDBP, hnRNPAB and hnRNPA2B1 and FUS itself [61]. Further, authors show that an increase in FUS causes a downregulation of PRMT5 expression leading to an autoregulatory accumulation of FUS, thus increasing the complexity of this regulatory mechanism [61].

Wu and coworkers investigated the functions of the lncRNA *RMST* (RhabdoMyosarcoma-associated Transcript) that has been characterized as a tumor suppressor in triple-negative breast cancers as well as a regulator of neuronal differentiation and brain development [62]. Authors reported that FUS and *RMST* directly interact and *RMST* enhances FUS SUMOylation [62] but fails to provide a mechanistic explanation for the *RMST*-dependent FUS SUMOylation. *RMST*-induced SUMOylation is required for the interaction between FUS and hnRNPD that is able to affect the stability of ATG4D protein, a factor involved in the biogenesis of autophagosomes, vesicles that contain cellular material intended to be degraded by autophagy [62]. Altogether, these data suggest that *RMST*-dependent SUMOylation of FUS promotes the hnRNPD-mediated stabilization of ATG4D and potentially impacts on the autophagic process [62].

The lncRNA *OCC1* (Overexpressed in Colon Carcinoma-1) plays a tumor suppressive role in colorectal cancer [63]. *OCC1* knockdown promotes cell growth both in vitro and in vivo, which is largely due to its ability to inhibit G0 to G1 and G1 to S phase cell cycle transitions [63]. *OCC1* exerts its function by destabilizing ELAVL1 (a.k.a. HuR) an RBP that, by interacting with the 3′ untranslated regions of its target mRNAs, can stabilize thousands of transcripts [64]. *OCC1* enhances the binding of an ubiquitin E3 ligase to ELAVL1 and renders the RBP susceptible to ubiquitination and degradation, thereby reducing the levels of ELAVL1 and, in turn, of its target mRNAs, including the mRNAs associated with cancer cell growth [63]. This report confirms the original observation that ELAVL1 undergoes regulated ubiquitination and proteasome degradation [64] and represents an example of a lncRNA that indirectly regulates the stability of a group of mRNAs through modulation of the post-translational modification of an RBP [63].

As anticipated in Section 1, levels of *MALAT1* affect the ratio between dephosphorylated and phosphorylated SF SRSF1 with a not completely defined mechanism [17].

## 7. LncRNAs, RBPs, and Maturation of microRNAs from Precursors

A flood of studies published in the last 20 years have demonstrated that microRNAs (miRNAs) regulate the entire spectrum of cellular functions and a number of reports clearly demonstrated that miRNA biogenesis is an important regulatory step that controls the cellular levels of miRNAs and, consequently, their functions [65]. The biogenesis of miRNAs involves two distinct enzymatic reactions carried out by distinct multiprotein complexes located in different cellular compartments [65]. First, primary miRNAs (pri-miRNAs) are processed to precursor miRNAs (pre-miRNAs) through the intervention of the DROSHA-containing complex in the nucleus. Next, through the interaction with XPO5 (a.k.a. exportin-5) and RAN, the pre-miRNA is transported into the cytoplasm where it undergoes a second round of processing catalyzed by the DICER1-containing protein complex. Finally, one strand of the resulting short (21–25 nt) RNA duplex, that corresponds to the mature miRNA, is loaded into the RISC (RNA Induced Silencing Complex) to exert its mRNA targeting functions [65]. Numerous studies have demonstrated that specific RBPs associate with the enzymatic complexes responsible for miRNA maturation to provide specificity and/or to regulate their activity [65].

Groundbreaking investigations conducted in 2015 by Filipowicz laboratory demonstrated that, during the course of postnatal development of retinal photoreceptors, the accumulation of mature miR-183/96/182 is delayed compared with pri-miR-183/96/182 [66]. Authors identified the lncRNA *Rncr4* (named after Retinal Non-Coding RNA 4) that is expressed in maturing photoreceptors as a factor activating pri-miR-183/96/182 maturation [66]. *Rncr4* modulates the activity of the DEAD-box RNA helicase/ATPase DDX3X, an RBP that exerts a potent inhibition on pri-miR-183/96/182 maturation in early phases of postnatal photoreceptor development [66]. Authors observe that the photoreceptor-specific DDX3X silencing results in a significant decrease in pri-miRNAs and a strong increase in mature miR-183/96/182 levels in photoreceptors when compared with controls [66]. MiR-183/96/182 control the expression of CRB1 that is a component of the molecular scaffold involved in the formation and integrity of tight junctions between retinal glia and photoreceptors that controls proper development of polarity in the eye [66]. Altogether the study reveals that *Rncr4*-regulated timing of miR-183/96/182 maturation from precursors is an essential step for obtaining the even distribution of cells across retinal layers.

More recently, Portman and coworkers utilized a different model of organ development—sexual maturation in *Caenorabditis Elegans (C. Elegans)*—to prove the involvement of lncRNA-regulated miRNA maturation from precursors during development [67]. The *C. Elegans* RBP LIN-28, similarly to mammalian LIN28, is a negative regulator of the maturation of let-7 miRNA family members from their pri-miRNAs and Portman and coworkers demonstrated that the lncRNA *lep-5* inhibits LIN-28 function thus promoting the maturation of let-7 that, in turn, controls the onset of sexual maturation in the nervous system of roundworms [67]. Mechanistically, *lep-5* functions as an RNA scaffold, forming a tripartite complex with LEP-2 (whose mammalian homolog is MKRN1 an E3 ubiquitin ligase that promotes the ubiquitination and proteasomal degradation of target proteins) and LIN-28 to promote LIN-28 degradation [67]. The well-known conservation of regulatory mechanisms across species allowed Portman and coworkers to hypothesize that an unidentified *lep-5*-like lncRNA may exist in mammals and play a key role in sexual maturation [67].

The heterodimeric complex formed by the two RBPs NONO and SFQP (a.k.a. PSF) has been defined as a prototypical multipurpose molecular scaffold that dynamically mediates a wide range of protein–protein and protein–nucleic acid interactions [68]. Indeed, the NONO-SFQP complex (*i*) controls pre-mRNA splicing and polyadenylation processes [68], (*ii*) plays a role in nuclear retention of defective RNAs—when associated with the nuclear matrix protein MATR3—, and (*iii*) promotes DNA double-strand break repair via the canonical non-homologous end joining pathway [68]. Fu and coworkers reported an additional function for the NONO-SFQP complex by demonstrating its ability to bind to a large number of pri-miRNAs and to globally enhance pri-miRNA processing into pre-miRNAs by the DROSHA complex [69]. The NONO-SFQP heterodimer is involved in paraspeckle formation and integrity and, therefore, it is not surprising that it interacts with the paraspeckle-enriched lncRNA *Neat1*. The authors also prove that *Neat1* specifically links NONO-SFQP heterodimer with the DROSHA complex thus modulating its enzymatic activity [69].

As we have discussed in Section 3, the lncRNA *H19* is endowed with remarkably distinct regulatory properties. Wu and coworkers recently reported that *H19* suppresses the expression of PTBP1 in cholestatic mouse livers [70]. Authors have observed that PTBP1 and *H19* interact under normal conditions but fail to provide information about the mechanism by which *H19* controls PTBP1 expression [70]. It would be interesting to investigate whether *H19* exerts a scaffold function by bridging together a putative ubiquitin ligase with PTBP1 in order to promote its degradation similarly to what *lep-5* does with LIN-28 in *C. Elegans* (see above, [67]). Authors report a suppressive effect of PTBP1 on the maturation of let-7 family members from their pre-miRNA precursors and suggest that *H19*-dependent PTBP1 downregulation ultimately leads to enhanced levels of let-7 family members in cholestatic mouse livers [70].

Our laboratory has reported that *H19* is indirectly implicated in the processing of a specific subset of miRNAs, the so-called myogenic-miRNAs, whose enhanced expression contributes to myogenesis and muscle regeneration [44]. Indeed, during myogenic differentiation of multipotent mesenchymal C2C12 cells, AKT-dependent phosphorylation of the RBP KHSRP induces its dismissal from the cytoplasm (where it is associated with *H19* to promote decay of labile mRNAs including *Myog*, see Section 3) and its translocation to cell nuclei where KHSRP is repurposed to induce myogenic-pri-miRNAs maturation [44].

## 8. Take-Home Message

It is evident from the above Sections that the networks based on lncRNA-RBP interactions represent highly versatile tools to post-transcriptionally regulate gene expression. We have discussed examples of specific lncRNAs that, through interactions with distinct sets of RBPs, regulate complex layers of post-transcriptional control (Summarized in Figure 1 and Table 1).

LncRNAs usually display a cell- or tissue-restricted expression while RBPs are more broadly expressed. Thus, a lncRNA can provide a cell- and/or tissue-specific function to an RBP. Further, since the expression levels of lncRNAs can be modulated by extracellular signals and RBP functions can be post-translationally modulated by the same and/or different pathways, the functional outcome of lncRNA-RBP complexes can be tightly controlled in a time- and space-specific manner. This results in a huge regulatory potential.

It is known that many lncRNAs function as molecular decoys and we have reviewed examples of abundant lncRNAs that exert part of their biological functions through this mechanism (e.g., *MALAT1, NEAT1, H19, NORAD*). However, the generally low abundance of many lncRNAs can generate debate on the stoichiometry of their interaction with the usually abundant RBPs. More and more evidence points to the functional relevance of specialized membrane-free subcellular compartments where high abundance of lncRNAs may not be required because their local concentration might be the limiting step. Indeed, ncRNAs have been viewed as potential mediators of liquid–liquid phase separation through their ability to operate as molecular scaffolds for the binding of RBPs, thus regulating the sizes and the dynamics of membrane-free organelles that carry out biological processes [71]. Phase separation is an emerging paradigm for understanding spatial and temporal regulation of a variety of cellular processes and additional studies will be needed to clarify its role in the post-transcriptional regulatory layer of gene expression [72].

In conclusion, the complexity of lncRNA-RBP functional networks is often increased by the experimental evidence that some post-transcriptional modifications of gene expression occur co-transcriptionally and by the ability of some lncRNAs to exert both transcriptional and post-transcriptional functions in a coordinated way. Recently developed technologies aimed at analyzing —in the context of distinct cell compartments—macromolecular complexes including lncRNAs, chromatin, and RBPs in an “almost-native” status, will allow researchers to portray, at a better resolution, the elaborate scenario of the interactions that we have described.

## Figures and Tables

**Figure 1 ncrna-06-00040-f001:**
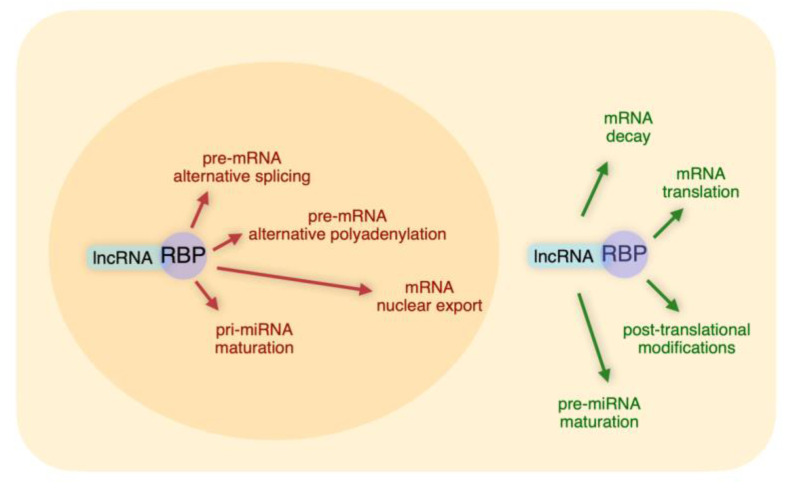
Nuclear and cytoplasmic functions of long non-coding RNA (lncRNA)-RNA binding protein (RBP) networks.

**Table 1 ncrna-06-00040-t001:** Summary of the lncRNA-RBP networks described in this review. The ENSEMBL accession number is provided in parentheses. In the case of *Drosophila* and *C. Elegans* lncRNAs, the accession numbers to FlyBase and WormBase, respectively, are provided in parentheses.

LncRNA	RBP	Function	Mechanism of Action	Ref.	Cell Outcome
*MALAT1* (ENSG00000251562)	Multiple splicing regulators	Alternative Splicing	Scaffold	[16,17,18,19,20,21]	Various
	YTHDC1, SRSF3	Nuclear export	Scaffold	[37]	Oncogenesis
	SRSF1	Alternative Splicing	Scaffold	[17]	Unknown
*NEAT1* (hsa ENSG00000245532) (mmu ENSMUSG00000092274)	Multiple splicing regulators	Alternative Splicing	Scaffold	[20,22,23,24]	Various
	NONO, SFQP	Pri-miRNA processing	Scaffold	[70]	Myoblast differentiation
*Miat* (ENSMUSG00000097767)	Multiple splicing regulators	Alternative Splicing	Scaffold	[25]	Control of neuronal depolarization
*PANDAR* (ENSG00000281450)	PTBP1	Alternative Splicing	Decoy	[26]	Apoptosis
*Pnky* (ENSMUSG00000107859)	PTBP1	Alternative Splicing	Unknown	[27]	Neurogenesis
*LINC01133* (ENSG00000224259	SRSF6	Alternative Splicing	Decoy	[28]	Epithelial to mesenchymal transition
*DSCAM-AS1* (ENSG00000235123)	hnRNPL	Alternative Splicing, Alternative polyadenylation	Scaffold	[30]	Cancer progression
*ROCR* (ENSG00000228639)	unknown	Nuclear export	Unknown	[34]	Cancer progression
*NORAD* (ENSG00000260032)	PUMILIO	mRNA decay	Decoy	[39,40,41,42]	Genome stability
*H19* (hsa ENSG00000130600) (mmu ENSMUSG00000000031)	KHSRP	mRNA decay	Scaffold	[44]	Myoblast differentiation
	Phospho-KHSRP	Pri-miRNA processing	Release of scaffold function	[44]	Myoblast differentiation
	PTBP1	Pre-miRNA processing	Indirect regulation?	[71]	Liver disease
*Epr* (ENSMUSG00000074300)	KHSRP	mRNA decay	Decoy	[45]	Cell proliferation
*LERFS* (ENSG00000234665)	SYNCRIP	mRNA decay/mRNA translation	Scaffold	[47]	Synoviocyte proliferation and motility
*UCA1* (ENSG00000214049)	hnRNPA1	mRNA decay	Decoy	[48]	Cell proliferation, senescence
	PTBP1	mRNA translation	Decoy	[55]	Cell proliferation
*MY* (ENSG00000261373)	hnRNPK	mRNA decay	Scaffold	[49]	Cell proliferation
*LINC-ROR* (ENSG00000258609)	PTBP1, hnRNPD	mRNA decay	Scaffold Decoy	[50]	Cell proliferation
	Phospho-PTBP1	mRNA translation	Decoy	[54]	Cell proliferation, apoptosis
*LncMyoD* (ENSMUST00000209655)	IGF2BP2	mRNA translation	Decoy	[56]	Myoblast differentiation
*LncSMaRT* (ENSMUSG00000087591)	DHX36	mRNA translation	Decoy	[57]	Myoblast differentiation
*BCYRN1* (ENSG00000236824)	hnRNPE1/E2	mRNA translation	Decoy	[58]	Post-synaptic translation
*HSRω (Drosophila)* (FlyBase ID FBgn0001234)	FUS	Post-translation modification	Indirect regulation	[61]	Neurotoxicity
*RMST* (ENSG00000255794)	FUS	Post-translation modification	Scaffold?	[62]	Autophagy
*OCC1* (ENSG000002351629	ELAVL1	Post-translation modification	Scaffold	[63]	Cell proliferation
*Rncr4* (ENSMUSG00000103108)	DDX3X	Pri-miRNA processing	Decoy	[66]	Photoreceptor development
*Lep-5* (*C. Elegans*) (WormBase H36L18.2)	LIN-28	Pri-miRNA processing	Scaffold	[67]	Sexual development

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
