# Peer review of "Long Non-Coding RNA-Ribonucleoprotein Networks in the Post-Transcriptional Control of Gene Expression"

_ncrna, 2020, doi:10.3390/ncrna6030040_

Round 1

Reviewer 1 Report

Briata and Gherzi have reviewed complexes comprising long noncoding RNAs and RNA-binding proteins RNA-binding proteins (RBPs) and lncRNAs.  They review lncRNPs influencing gene expression programs by modulating splicing, polyadenylation, mRNA export, mRNA turnover, translation, protein stability, and microRNA maturation from precursor RNAs.

The authors have extensive expertise in this field and have prepared an interesting and timely review.  To improve the readability, the authors are encouraged to carry out the following changes:

  • The piece needs to be edited for typos and grammatical errors. Moreover, to follow convention, lncRNAs should be written in italics (all caps for human, all lowercase except the first letter for mouse); mRNAs should follow the same rule.
  • Minor point: the categories of lncRNAs listed on page 1 should be completed. Besdies those in intergenic regions, antisense RNAs, and intronic RNAs, there are lncRNAs expressed from regular exons and may even undergo splicing.
  • A point of controversy in the area of lncRNP biology is stoichiometry, i.e., the relative abundance of lncRNAs and RBPs. I would be interested in having the authors address this issue, as RBPs tend to be quite abundant, while lncRNAs are generally in low abundance.  How do the authors envision spatial distributions or RBP availability that enable functional lncRNA-RBP interactions?  Are there other considerations that explain how disparate concentrations of RBPs and lncRNAs result in complexes with functional meaning?
  • I strongly urge the authors to have one or several figures that illustrate the functions of lncRNPs described in the text.
  • Some sections are quite long. The creation of subsections for those that are longest (for example, sections 2 and 4) would improve readability.

Author Response

We thank the Reviewer for the appreciation of our manuscript and for her/his useful criticisms.

The piece needs to be edited for typos and grammatical errors. Moreover, to follow convention, lncRNAs should be written in italics (all caps for human, all lowercase except the first letter for mouse); mRNAs should follow the same rule.

We have carefully edited the text in order to amend typos and grammatical errors. We have also improved the style of some Sections (e.g. the Abstract). Further, we have written in the appropriate style the names of lncRNA and mRNA molecules. All changes are highlighted in yellow.

Minor point: the categories of lncRNAs listed on page 1 should be completed. Besdies those in intergenic regions, antisense RNAs, and intronic RNAs, there are lncRNAs expressed from regular exons and may even undergo splicing.

We have changed the sentence according to the Reviewer’s suggestion (please see lines 30-32 in the revised manuscript.

A point of controversy in the area of lncRNP biology is stoichiometry, i.e., the relative abundance of lncRNAs and RBPs. I would be interested in having the authors address this issue, as RBPs tend to be quite abundant, while lncRNAs are generally in low abundance. How do the authors envision spatial distributions or RBP availability that enable functional lncRNA-RBP interactions?  Are there other considerations that explain how disparate concentrations of RBPs and lncRNAs result in complexes with functional meaning?

We have discussed this interesting point at lines 504-515 and added two new references.

I strongly urge the authors to have one or several figures that illustrate the functions of lncRNPs described in the text.

We added a Figure that summarizes the mode of action of lncRNA-RBP complexes in the post-transcriptional regulation of gene expression.

Some sections are quite long. The creation of subsections for those that are longest (for example, sections 2 and 4) would improve readability.

Unfortunately, it is impossible to create sub-section because there is no conceptual separation of topics. However, we made a format change in order to separate the general introduction of each Section from the description of the specific lncRNAs.

Reviewer 2 Report

The article is very relevant given the accumulation of recent data on the interaction of lncRNAs with RBPs forming multifunctional complexes. However, I strongly suggest the inclusion of figures that represent the main lncRNA-RBP networks.

As lncRNAs still not have a standardized nomenclature, I suggest that the authors include as a new column in table, the Ensemble/Gene ID number for each one.

Line 83 - As you will not comment about the other types of regulation by lncRNAs , I suggest some references to point them.

Author Response

The article is very relevant given the accumulation of recent data on the interaction of lncRNAs with RBPs forming multifunctional complexes. However, I strongly suggest the inclusion of figures that represent the main lncRNA-RBP networks.

We thank the Reviewer for the appreciation of our manuscript and for her/his useful criticisms. According to Reviewer’s suggestions, we added a Figure that summarizes the mode of action of lncRNA-RBP complexes in the post-transcriptional regulation of gene expression.

As lncRNAs still not have a standardized nomenclature, I suggest that the authors include as a new column in table, the Ensemble/Gene ID number for each one.

We added in the first column of the Table the ENSEMBL accession number for human and mouse lncRNAs and, in the case of Drosophila and C. Elegans lncRNAs, the accession numbers to FlyBase and WormBase, respectively.

Line 83 - As you will not comment about the other types of regulation by lncRNAs, I suggest some references to point them.

According to Reviewer’s suggestion, we have now referenced this sentence (lines 83, 84 in the revised version).

Reviewer 3 Report

Briata and Gherzi put together a small, yet exhaustive, review on the current knowledge of RNP complexes composed by lncRNAs and RBP. 

This reviewer finds the review worthy of publication w/o any extensive review.

The only suggested modification is about the sentence in lane 61; specifically, splicing and termination happen in a co-transcriptional fashion. Thus, authors should write “…co- and post-transcriptional…”

Best Regards

Author Response

This reviewer finds the review worthy of publication w/o any extensive review.

We thank the Reviewer for the appreciation of our manuscript and for her/his useful suggestion.

The only suggested modification is about the sentence in lane 61; specifically, splicing and termination happen in a co-transcriptional fashion. Thus, authors should write “…co- and post-transcriptional…”

According to Reviewer’s suggestion we have added a new sentence (lines 79 and 80) in the Section where we discuss co-transcriptional modifications of nascent transcripts.